# Improving formaldehyde consumption drives methanol assimilation in engineered *E. coli*

Benjamin M. Woolston[1], Jason R. King[1,3], Michael Reiter[1], Bob Van Hove[2] & Gregory Stephanopoulos[1]

Due to volatile sugar prices, the food vs fuel debate, and recent increases in the supply of natural gas, methanol has emerged as a promising feedstock for the bio-based economy. However, attempts to engineer *Escherichia coli* to metabolize methanol have achieved limited success. Here, we provide a rigorous systematic analysis of several potential pathway bottlenecks. We show that regeneration of ribulose 5-phosphate in *E. coli* is insufficient to sustain methanol assimilation, and overcome this by activating the sedoheptulose bisphosphatase variant of the ribulose monophosphate pathway. By leveraging the kinetic isotope effect associated with deuterated methanol as a chemical probe, we further demonstrate that under these conditions overall pathway flux is kinetically limited by methanol dehydrogenase. Finally, we identify NADH as a potent kinetic inhibitor of this enzyme. These results provide direction for future engineering strategies to improve methanol utilization, and underscore the value of chemical biology methodologies in metabolic engineering.

---

[1] Department of Chemical Engineering, Massachusetts Institute of Technology, 25 Ames Street, MIT 56-469C, Cambridge, MA 02139, USA. [2] Centre for Synthetic Biology (CSB), Department of Biochemical and Microbial Technology, Ghent University, 9000 Ghent, Belgium. [3]Present address: Department of Organism Engineering, Ginkgo Bioworks, 27 Drydock Ave, Suite 800, Boston, MA 02210, USA. Correspondence and requests for materials should be addressed to G.S. (email: gregstep@mit.edu)

Aprinciple motivation for industrial biomanufacturing is the desire to replace fossil fuel-based processes with more sustainable ones utilizing renewable feedstocks[1]. While most efforts to date have relied on easily accessible sugars from corn or sugarcane, these substrates compete with food supply. In the quest for alternative feedstocks, methanol has emerged as a promising candidate[2]. Its current price is comparable to that of glucose, despite being more energy-dense, with a 50% greater degree of reduction per C-mole. Furthermore, methanol can be produced renewably from municipal solid waste and biogas through syngas as an intermediate[3], or by using renewable electricity through electrocatalytic reduction of $CO_2$[4]. Conversion of methane to methanol as a starting material for value-added processes has also been proposed as a way to monetize stranded natural gas, a large fraction of which is currently flared in the absence of an economically viable alternative[5,6]. For these reasons, the concept of a methanol economy has received considerable attention[7].

The use of methanol in bioprocessing presents several challenges. Chief among these is that there has been only limited success in engineering methylotrophs to produce high levels of heterologous products[2,8–10]. Metabolic engineering in methylotrophs is relatively slow, and the genetic tools are not as well developed as for model organisms. For example, targeted gene deletions in Bacillus methanolicus, a methylotroph of commercial interest for its native production of amino acids, have not been reported to date. Because of this, significant interest has arisen in importing methanol assimilation into the more tractable host, Escherichia coli. This concept, dubbed synthetic methylotrophy[11], aims to combine the advantages of methanol as an attractive carbon substrate for fermentation with the robust engineering tools and range of downstream heterologous products available in E. coli.

Several groups have attempted to develop a methylotrophic E. coli. In the seminal paper, Müller and co-workers demonstrated the ability of E. coli cells expressing a heterologous operon containing methanol dehydrogenase (Mdh) and the ribulose monophosphate (RuMP) pathway enzymes hexulose phosphate synthase (Hps) and phosphohexulose isomerase (Phi) to incorporate [13]C methanol into central carbon metabolites[12]. Recently, Whitaker and co-workers showed that supplementation of methanol during growth on yeast extract could allow a similarly engineered strain to reach higher cell densities than with yeast extract alone, and could produce the heterologous product naringenin with a significant amount of the carbon derived from methanol[13]. Follow-up work found threonine to be the primary beneficial component of yeast extract[14], and re-routing glucose flux through the oxidative pentose phosphate pathway (PPP) by knocking out pgi further improved methanol assimilation[15]. To date, however, no group has been able to demonstrate growth of E. coli on methanol as a sole carbon source. Similar efforts in Corynebacterium glutamicum have met with the same results[16].

Various hypotheses of potential bottlenecks to methanol assimilation have been put forth. Chief among these are the relatively poor kinetics of heterologously expressed Mdh and the thermodynamically challenging oxidation of methanol to formaldehyde with NAD as electron acceptor. Low activity of the enzyme limits total methanol flux, and the high thermodynamic barrier requires both a high intracellular NAD:NADH ratio and rapid consumption of formaldehyde through the ribulose monophosphate (RuMP) cycle, to keep reaction far from equilibrium and driven in the forward direction. This has led to efforts to find and engineer more active variants of Mdh[17], and a strategy involving the co-localization of Mdh and Hps, the first enzyme of the RuMP pathway, to prevent the build-up of formaldehyde[18]. However, the effects of these modifications on the incorporation of methanol into central metabolism have not been characterized. More broadly, there has been no systematic attempt to quantitatively assess the potential bottlenecks of methanol assimilation. Such an analysis is a critical component of developing a rational strategy to improve methanol assimilation, because it justifies the considerable time and resource expenditure associated with developing a solution to the proposed bottleneck.

To address this need, in this work we engineer strains with similar performance to the previously reported strains, and develop quantitative assays to systematically assess pathway limitations to identify the most promising targets for further metabolic engineering. Our analysis reveals the depletion of ribulose 5-phosphate (Ru5P), the co-substrate of Hps, as the primary limitation to high methanol flux in carbon-starved cells, as this leads to the build-up of formaldehyde and the dissipation of the driving force. By inhibiting glycolytic flux with iodoacetate and overexpressing E. coli glpX, we activate the sedoheptulose bisphosphatase (SBPase) variant of the RuMP pathway, which reduces the steady-state formaldehyde concentration by a factor of three, and increases the incorporation of [13]$CH_3OH$ by a factor of two. Thermodynamic calculations and experiments using the kinetic isotope effect associated with deuterated methanol show that, under these conditions, methanol oxidation is forward driven and limited by the kinetics of Mdh. These results represent a significant step toward establishing synthetic methylotrophy, and provide specific direction for subsequent work to develop a methylotrophic E. coli.

## Results

**Formaldehyde assimilation is limited in carbon-starved cells.** The engineered pathway of methanol assimilation is shown in Fig. 1. Overcoming the uphill thermodynamics of NAD-dependent methanol oxidation ($\Delta G^{0'} = +34.2$ kJ mol$^{-1}$) requires a low concentration of the product formaldehyde ($CH_2O$) to maintain the driving force. With a [$CH_3OH$] of 250 mM and a typical E. coli NAD:NADH ratio of 31.3[19], the reaction is predicted to reach equilibrium at a [$CH_2O$] of just 50 μM (Supplementary Fig. 1, Supplementary Note 1, and Supplementary Table 1). Rapid formaldehyde assimilation by the downstream pathway enzymes is therefore necessary for maintaining forward-driven methanol assimilation.

To examine the ability of E. coli cells engineered for methanol metabolism to metabolize formaldehyde, we used the assay procedure developed by Müller et al.[12] to measure [$CH_2O$] in carbon-starved cells treated with 250 mM methanol. Starved cells were chosen in light of the eventual goal of developing a strain that can grow solely on methanol, which requires that this strain must be able to initiate and sustain methanol consumption in the absence of any other carbon substrate. Cells expressing the evolved Mdh from C. necator[17] (plasmid pETmdh) produced 56 ± 4 μM formaldehyde, with an initial rate of 7.2 ± 0.2 μM min$^{-1}$ OD$^{-1}$ (intervals represent s.d., $n = 3$, Fig. 2), which is similar to those achieved previously[12,13]. Surprisingly, cells additionally expressing Hps from B. methanolicus and Phi from Methylococcus capsulatus (plasmid pETMEOH500) produced only slightly less formaldehyde (Fig. 2b), despite the superior kinetics of these enzymes compared to Mdh (Supplementary Table 2). We hypothesized that availability of Ru5P, the co-substrate of Hps, might limit formaldehyde assimilation, and performed the assay again while supplementing 6 g L$^{-1}$ xylose, which is metabolized through the pentose phosphate pathway (PPP, Fig. 2a), to increase the [Ru5P]. This resulted in a dramatically reduced steady-state [$CH_2O$] of 7.5 ± 2.6 μM or 4.7 ± 1.7-fold compared to the Mdh control (Fig. 2b). To confirm that the reduction in [$CH_2O$] was due to increased availability of Ru5P, we measured

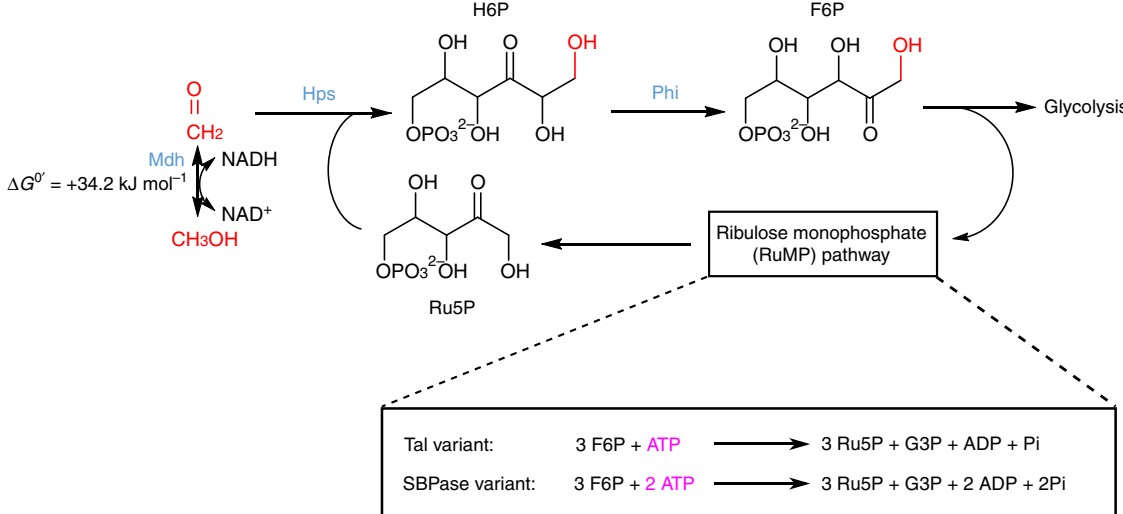

**Fig. 1** The pathway of methanol assimilation in *E. coli*. Heterologous enzymes required for methanol assimilation in *E. coli* are shown in blue. Methanol dehydrogenase (Mdh) catalyzes the thermodynamically uphill NAD-dependent oxidation of methanol to formaldehyde ($\Delta G^{0'} = +34.2 \text{ kJ mol}^{-1}$), which is ligated to ribulose 5-phosphate (Ru5P) in an aldol reaction catalyzed by hexulose phosphate synthase (Hps). The resulting hexulose (H6P) is isomerized by phosphohexulose isomerase (Phi) to fructose 6-phosphate (F6P), which is metabolized through native *E. coli* central metabolism. A portion of the F6P must be used to regenerate Ru5P through the ribulose monophosphate (RuMP) cycle to enable further formaldehyde assimilation. Two major variants of this cycle are known, the transaldolase (Tal) and sedoheptulose bisphosphatase (SBPase), which are named for the enzyme that produces sedoheptulose 7-phosphate, and differ in their ATP requirements. The red moiety denotes the position of assimilated methanol. H6P: D-arabino-3-hexulo-6-phosphate, G3P: glyceraldehyde 3-phosphate

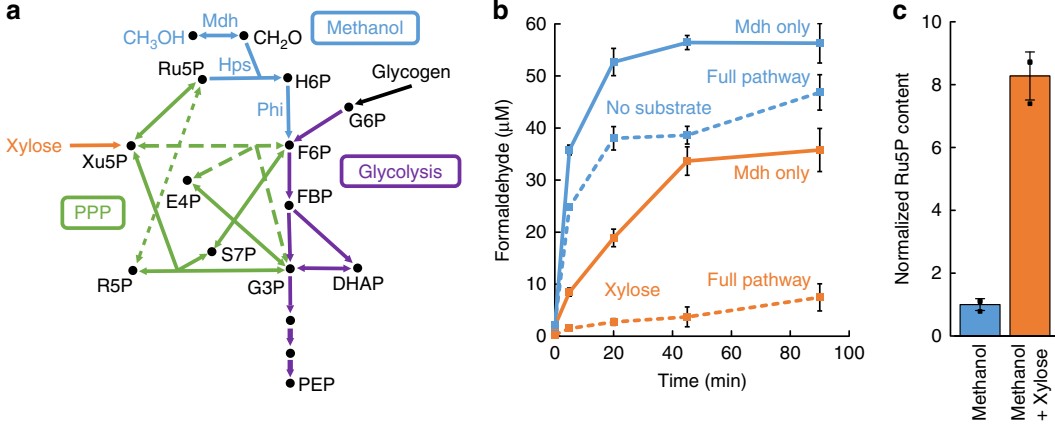

**Fig. 2** Insufficient Ru5P concentration in carbon-starved cells limits formaldehyde assimilation. **a** Detailed enzymatic reactions and metabolic pathways involved in methanol metabolism, showing the entry points of glycogen (the primary carbon source during starvation) through glycolysis, and xylose through the pentose phosphate pathway (PPP). Heterologous reactions unique to methanol assimilation are shown in blue. Glycolysis reactions are shown in purple, and PPP reactions in green. Dotted and dashed green lines added for clarity. **b** Formaldehyde levels over time after addition of 250 mM methanol to starved cells of *E. coli* MG1655(DE3) $\Delta frmA$ with either no additional substrate (blue) or 6 g L$^{-1}$ xylose (orange). Solid lines represent cells expressing only Mdh, and dashed lines denote cells expressing the full methanol assimilation pathway (Mdh, Hps, and Phi). **c** Relative ribulose 5-phosphate (Ru5P) concentration in cells with no substrate or xylose. Error bars represent s.d. of $n = 3$ biological replicates (three individual colonies). H6P: D-arabino-3-hexulo-6-phosphate, F6P: fructose 6-phosphate, G6P: glucose 6-phosphate, FBP: fructose 1,6-bisphosphate, DHAP: dihydroxyacetone phosphate, G3P: glyceraldehyde 3-phosphate, PEP: phosphoenolpyruvate, E4P: erythrose 4-phosphate, S7P: sedoheptulose 7-phosphate, R5P: ribose 5-phosphate, Ru5P: ribulose 5-phosphate, Xu5P: xylulose 5-phosphate

the intracellular [Ru5P] in cells with and without xylose supplementation, and found an 8.2 ± 1.7-fold increase in the presence of added xylose (Fig. 2c). Additionally, repeating the experiment with glycerol as supplemental carbon source resulted in a much less significant reduction in [CH2O], indicating the effect was not simply due to a general response of starved cells to a growth substrate (Supplementary Fig. 2).

Interestingly, cells expressing only Mdh produced less formaldehyde when supplemented with xylose than the unsupplemented control, and at a slower initial rate (Fig. 2b). The lower steady state is likely due to the 3.2 ± 1.0-fold increased [NADH] in xylose-metabolizing cells compared to starved cells (Supplementary Fig. 3a), which again highlights the thermodynamic challenge associated with the reaction. The lower initial velocity suggested that NADH could be a kinetic inhibitor of the forward direction. This was verified by in vitro assay with purified Mdh, which showed ~50% reduction in activity in the presence of typical cellular levels of NADH (Supplementary Fig. 3b). These observations suggest that to further improve methanol oxidation rates, strategies should be developed to reduce the cellular [NADH], or engineer or evolve an NADH-insensitive variant of Mdh.

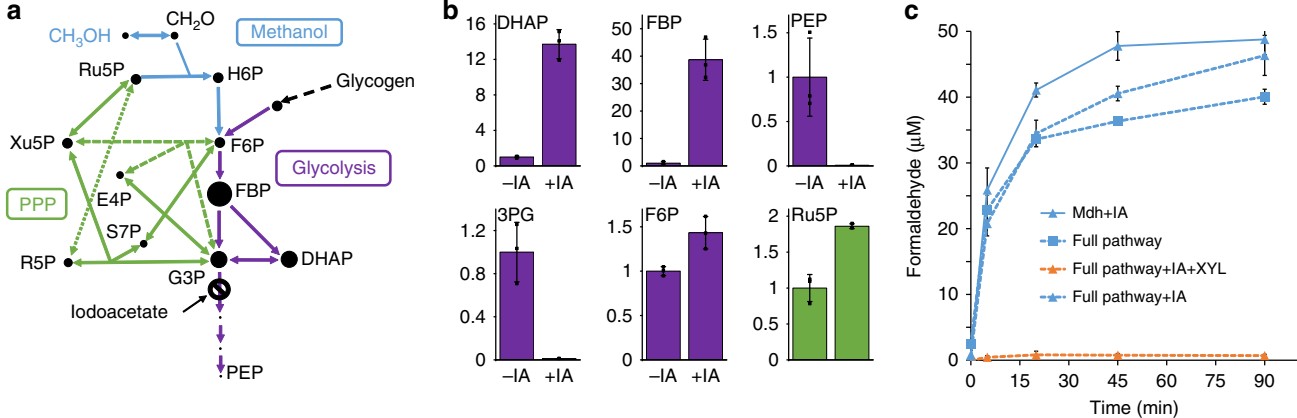

**Fig. 3** Chemical inhibition of glycolysis with iodoacetate is insufficient to improve formaldehyde assimilation. **a** Metabolic pathways in methanol metabolism, additionally depicting chemical inhibition of glyceraldehyde 3-phosphate dehydrogenase (GAPDH) by the addition of 1 mM iodoacetate (IA). Dot sizes indicate relative pool sizes after IA addition compared to untreated cells. **b** Relative internal concentrations of various metabolites with or without IA treatment, quantified by LC-MS/MS. Metabolites immediately upstream of GAPDH (DHAP and FBP) increased dramatically, while metabolites downstream of the blockage (e.g., PEP and 3PG) became virtually undetectable. **c** Formaldehyde levels after addition of 250 mM methanol to starved cells of *E. coli* MG1655(DE3) Δ*frmA*. Solid lines denote cells expressing only Mdh, whereas dashed lines indicate cells expressing the full pathway (Mdh, Hps, and Phi). Triangles indicate cells treated with IA. Blue lines symbolize cells supplemented with no additional substrate beyond methanol, whereas orange lines represent cells supplemented with 6 g L$^{-1}$ xylose (XYL). Error bars represent s.d. of $n = 3$ biological replicates (three individual colonies)

**The SBPase pathway improves formaldehyde assimilation.** Given the ultimate goal of using methanol as the sole carbon source for *E. coli*, we sought to improve the regeneration of Ru5P without the need for xylose supplementation. A genome-scale model of *E. coli* simulating hypothetical growth on methanol revealed a threefold decrease in proportional flux through glyceraldehyde 3-phosphate dehydrogenase (GAPDH) relative to the same flux for growth on glucose (Supplementary Fig. 4). This reaction connects the RuMP pathway to lower glycolysis (Fig. 3a), thus we reasoned that by inhibiting this reaction, we could maintain larger pool sizes of upper glycolytic and RuMP pathway intermediates in starved cells. To test this possibility experimentally, we repeated the assay in cells treated with 1 mM iodoacetate (IA), a potent inhibitor of GAPDH[20]. Inhibition was verified by analyzing metabolite pools by LC-MS/MS (Fig. 3b). Metabolites downstream of GAPDH, e.g., phosphoenolpyruvate (PEP) and 3-phosphoglycerate (3PG) became almost undetectable, whereas the concentrations of the metabolites directly upstream of GAPDH—G3P and DHAP—rose by a factor of 14 ± 2. Fructose bisphosphate (FBP), which is present in very low concentration in starved cells[21], increased in concentration by a factor of 39 ± 24. IA had no functional effect on the activity of Mdh, or on Hps and Phi, as evidenced by formaldehyde assay controls with cells expressing just Mdh, or with xylose supplementation (Fig. 3c). Despite the increases in [G3P] and [FBP], increases in [F6P] and [Ru5P] were comparatively small (Fig. 3b). Consequently, there was little improvement in [CH$_2$O] in cells expressing the full pathway ([CH$_2$O] = 46 ± 3 μM), compared to cells expressing only Mdh ([CH$_2$O] = 49 ± 2 μM) (Fig. 3c).

Entry into the transaldolase variant of the RuMP pathway in *E. coli* is mediated by the transaldolase and transketolase reactions (Fig. 4a), both of which require F6P as a substrate. Thus, in the absence of a large increase in [F6P], the lack of substantial increase in [Ru5P] upon IA treatment is not surprising. We therefore asked if we could convert the large increase in [FBP] upon IA treatment into an increase in [F6P]. F6P is converted to FBP by phosphofructokinase in an essentially irreversible reaction that uses ATP (Fig. 4b). We hypothesized that by expressing a gluconeogenic FBPase, which catalyzes the dephosphorylation of FBP to F6P with water, we could enable reverse carbon flow from FBP to upper glycolysis, and from there into the RuMP pathway.

*E. coli* has a major (Type I) and minor (Type II) FBPase, encoded by *fbp* and *glpX*, respectively. The major FBPase is inhibited by AMP[22]. Since [AMP] is increased in starved cells[21], we chose to overexpress the AMP-insensitive variant encoded by *glpX*[23] on a compatible plasmid (pACglpX), and repeat the formaldehyde assays.

Overexpression of *glpX* in combination with IA treatment resulted in a dramatic 6.4 ± 2.1-fold increase in [F6P], and a corresponding 4.0 ± 1.2-fold increase in [Ru5P] (Fig. 4c). Gratifyingly, this was accompanied by a 3.0 ± 0.3-fold decrease in steady-state [CH$_2$O] in cells expressing the full pathway ([CH$_2$O] = 17 ± 1.5 μM), compared to Mdh-only ([CH$_2$O] = 52 ± 1 μM) (Fig. 4d). Both IA and *glpX* overexpression were required to achieve this effect. To verify the reduction in [CH$_2$O] was due to increased assimilation into central metabolism, the experiment was repeated with $^{13}$CH$_3$OH, and the isotopic enrichment of various glycolytic intermediates measured. Labeling of the representative molecule F6P was approximately double in the IA + *glpX* treatment compared to the IA-only control treatment (Fig. 4e), with a mean isotopic enrichment of 27.5 ± 2.2% compared to 15.2 ± 0.5%, reflecting an increase in the rate of formaldehyde assimilation due to *glpX* overexpression. The increased labeling also confirms that Ru5P is not regenerated through the oxidative PPP, since this pathway results in the loss of labeled carbon. Further labeling experiments in a Δ*zwf* strain confirmed that there is little oxidative PPP flux under these conditions (Supplementary Fig. 5).

Some Type II FBPases are known to have promiscuous activity as sedoheptulose bisphosphatases (SBPases), including the enzyme encoded by plasmid-borne *glpX* in *B. methanolicus*[24]. If the *E. coli* variant was similarly promiscuous, conversion of SBP to S7P in the *glpX*-overexpressing strain would provide an alternative explanation for the improved formaldehyde assimilation (Fig. 5a). To test this possibility, we purified the *E. coli* GlpX, and assayed the conversion of SBP to S7P using LC-MS/MS in a coupled assay with purified fructose bisphosphate aldolase (FbaA), which promiscuously catalyzes the conversion of DHAP and E4P to the commercially unavailable SBP. In contrast to previous literature[23], we found that *E. coli* GlpX catalyzed the quantitative conversion of E4P and DHAP to S7P in the presence of FbaA (Fig. 5b), demonstrating this enzyme has SBPase activity,

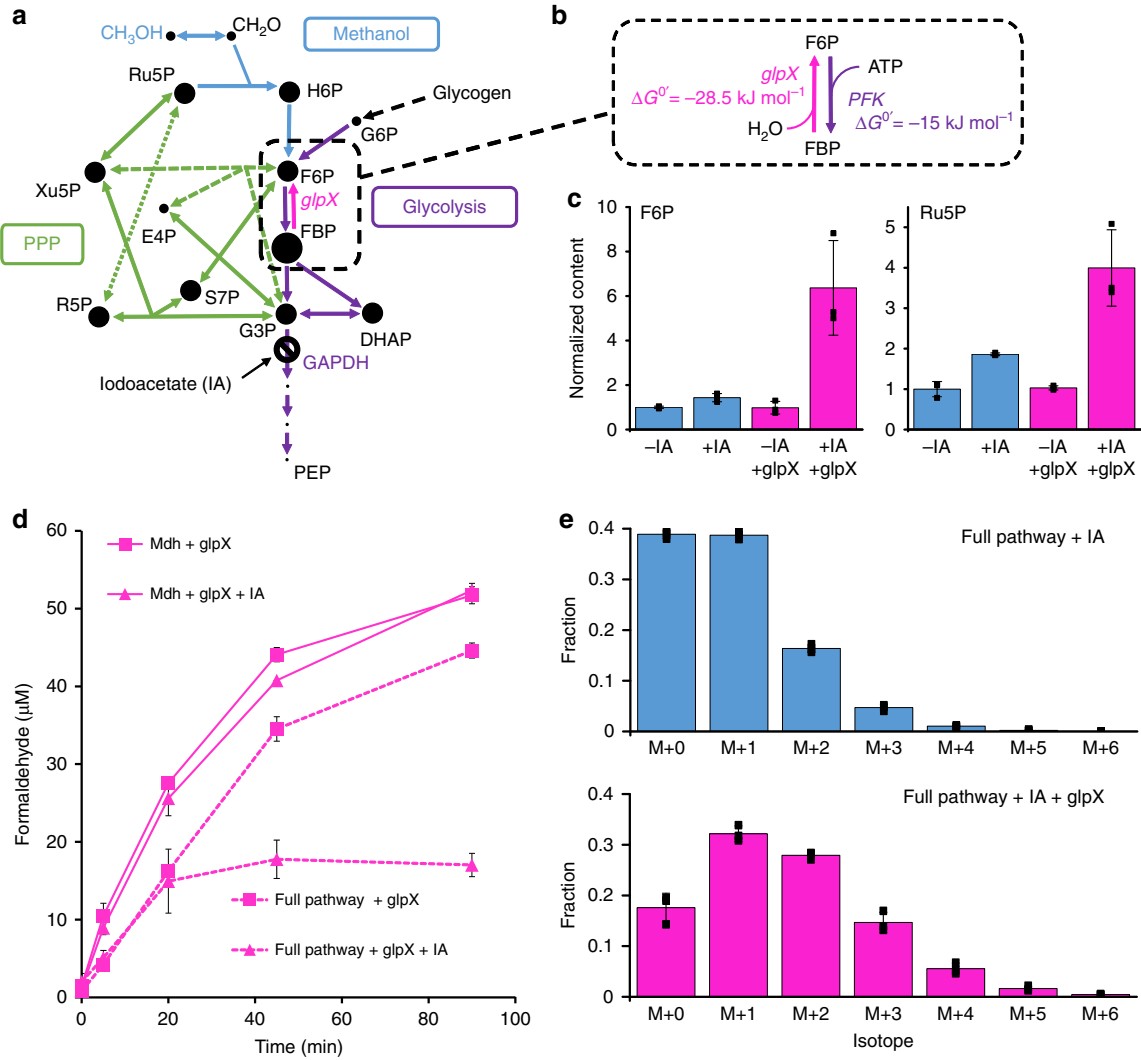

**Fig. 4** Iodoacetate coupled with *glpX* overexpression increases Ru5P concentration and improves cyclic formaldehyde assimilation. **a** Metabolic pathways in methanol metabolism, showing the blockage of glycolysis by iodoacetate (IA), and the minor gluconeogenic FBPase encoded by *glpX*. Dot sizes indicate relative pool sizes after IA addition in cells expressing *glpX* compared to untreated cells without *glpX* overexpression. **b** FBPase catalyzes the irreversible hydrolysis of FBP to F6P ($\Delta G^{0'} = -28.5$ kJ mol$^{-1}$), potentially allowing the conversion of the large FBP pool resulting from IA-mediated inhibition into F6P, the entry point for the transaldolase and transketolase reactions that replenish the PPP intermediates. **c** Relative concentrations of F6P and Ru5P upon treatment with IA in cells with (pink) or without (blue) *glpX* overexpression. **d** Formaldehyde levels after addition of 250 mM methanol to starved cells of *E. coli* MG1655(DE3) Δ*frmA*. Solid lines denote cells expressing only Mdh, whereas dashed lines indicate cells expressing the full pathway (Mdh, Hps, and Phi). Pink lines denote cells expressing *glpX*. Triangles indicate cells treated with IA, squares indicate untreated cells. **e** Isotopic analysis of fructose 6-phosphate extracted from IA-treated cells with (pink) or without (blue) *glpX* overexpression that were treated with 250 mM $^{13}$CH$_3$OH. Error bars represent s.d. of $n = 3$ biological replicates (three individual colonies)

and suggesting the possibility of carbon flux through the SBPase variant of the RuMP pathway. In this variant of the pathway, S7P is formed from through the dephosphorylation of SBP, which is in turn formed from erythrose 4-phosphate (E4P) and DHAP through FbaA (Fig. 5a, orange lines). This pathway requires an additional ATP compared to the transaldolase (TAL) pathway[25], in which S7P is formed from F6P and GAP (Fig. 5a, gray dotted lines). Analysis of metabolite pool sizes showed that, upon addition of IA, the [SBP] increased by $5.6 \pm 2.0$-fold. Over-expression of *glpX* reduced this to untreated levels, providing evidence of in vivo conversion of SBP to S7P (Fig. 5c). In addition, the [S7P] increased dramatically upon overexpression of *glpX*, regardless of whether cells were treated with IA or not ($8.3 \pm 1.8$-fold and $15 \pm 3$-fold, respectively, Fig. 5c).

In principle, increased [S7P] could be due to SBPase activity, or to the originally hypothesized increased entry into the RuMP pathway through transaldolase mediated by a higher [F6P]. To determine the relative contribution of the transaldolase pathway and the SBPase pathway, we generated a strain deficient in transaldolase activity (Supplementary Fig. 6a,b), and repeated the formaldehyde assay. If Ru5P regeneration was mediated primarily by increased entry into the PPP through transaldolase upon IA treatment and *glpX* overexpression, then elimination of transaldolase should counteract the effect of these treatments on reducing [CH$_2$O]. By contrast, if Ru5P regeneration was mediated by the SBPase pathway, deletion of transaldolase should have no impact on formaldehyde assimilation. The results are shown in Fig. 5d. As before, without the combination of IA treatment and *glpX* overexpression, steady-state formaldehyde levels were around 55 μM. In the *glpX* + IA treatment, [CH$_2$O] dropped to $18 \pm 5$ μM, a $2.5 \pm 0.7$-fold reduction which is comparable to that seen in the transaldolase-positive strain (Fig. 5d). The [S7P] also

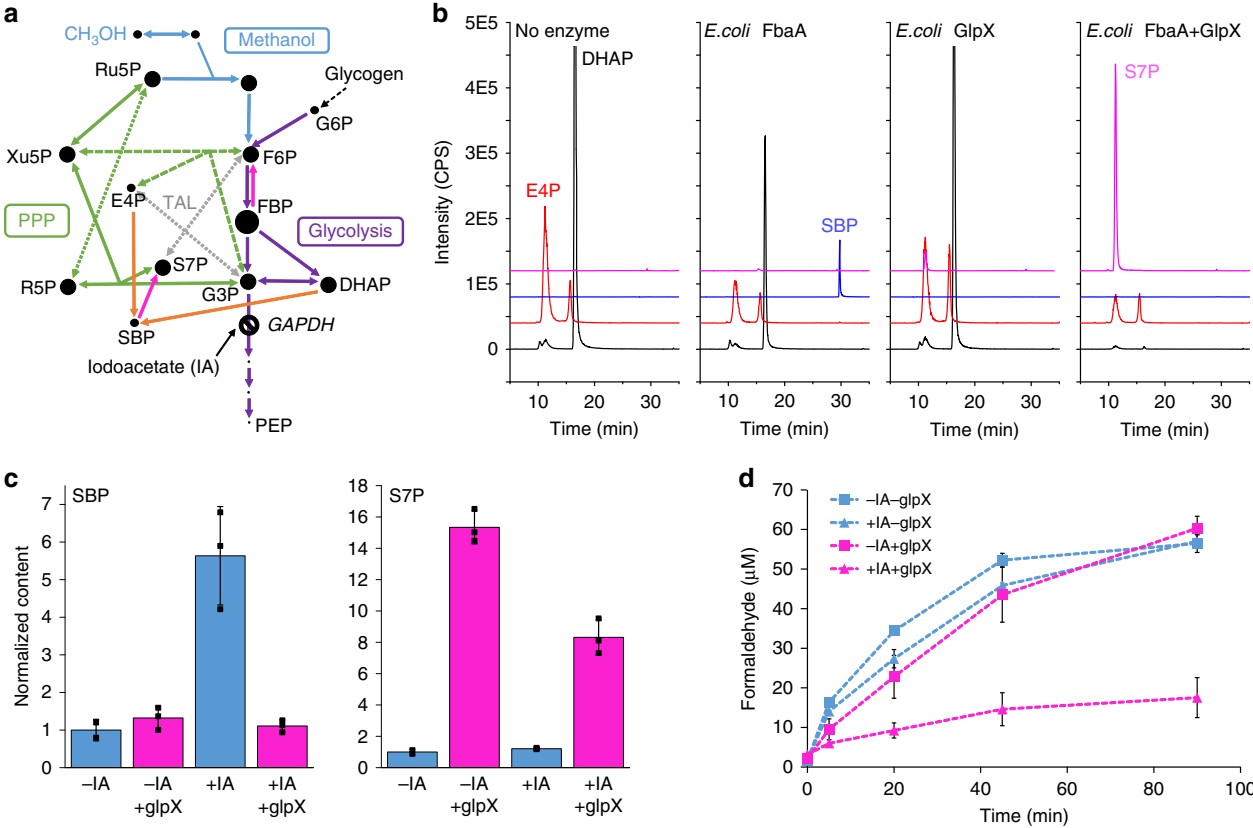

**Fig. 5** Ru5P regeneration is mediated by activation of the SBPase variant of the RuMP pathway. **a** Pathways involved in methanol metabolism, additionally showing the sedoheptulose bisphosphatase (SBPase) variant of the ribulose monophosphate (RuMP) pathway, where SBP is produced from DHAP and E4P by FbaA (orange line), and dephosphorylated to S7P by SBPase (pink line), compared to the transaldolase (TAL) pathway (gray line), where S7P is produced from F6P and E4P. Dot sizes indicate relative metabolite concentrations in cells treated with IA and overexpressing *glpX*, compared to untreated cells not overexpressing *glpX*. **b** LC-MS/MS analysis of assays with purified *E. coli* FbaA and GlpX, demonstrating that GlpX has SBPase activity. Black = DHAP, Red = E4P, Blue = SBP, Pink = S7P. **c** Relative SBP (left) and S7P (right) levels in cells treated with or without iodoacetate and expressing glpX (pink) or not (blue). **d** Formaldehyde timecourse in transaldolase-deficient *E. coli* cells (MG1655(DE3) ΔfrmA talA(K131A) ΔtalB) treated with 250 mM methanol, either expressing (pink) or not expressing (blue) glpX. Triangles denote cells treated with IA, and squares denote untreated cells. Error bars represent s.d. of *n* = 3 biological replicates (three individual colonies)

showed the same patterns as with the parent strain (Supplementary Fig. 6c). These data support the conclusion that the improved regeneration of Ru5P upon treatment with IA and overexpression of *glpX* is mediated by activation of the SBPase RuMP pathway.

**Assessing methanol dehydrogenase reversibility.** Having developed a strategy for achieving rapid consumption of formaldehyde without xylose supplementation, we sought to determine whether methanol oxidation was still thermodynamically limited under these conditions, as this would inform subsequent strategies to further improve methanol flux. Comparing the steady-state level of $CH_2O$ in strains expressing just Mdh to strains harboring the full pathway allows calculation of an upper limit for the $\Delta G$, and thus the ratio of forward to reverse Mdh flux[26] (Eq. 9), since the NAD:NADH ratio was the same in both cases (Supplementary Fig. 7). This analysis is shown in Fig. 6a. With no xylose supplementation, the small $1.2 \pm 0.1$-fold reduction in $[CH_2O]$ achieved by expression of the full pathway (Fig. 3b) means that only $55 \pm 3\%$ of the total flux can be guaranteed to be in the forward direction. With supplementation of xylose, a minimum of $83 \pm 5\%$ of the flux is in the forward direction, and with IA treatment coupled with *glpX* overexpression, $75 \pm 2\%$ of the flux is forward. This analysis suggests that, with a strategy to ensure sufficient Ru5P regeneration, the pathway is not limited by the thermodynamics of the Mdh reaction.

These calculations assume that there is no spatial variation of $[CH_2O]$. However, there is evidence that $[CH_2O]$ is higher in the immediate vicinity of the Mdh than in the bulk: in a recent study, co-localizing Mdh with Hps led to significant improved flux through the pathway[18]. We therefore sought an experiment to directly determine to what extent the Mdh reaction was thermodynamically limited. We reasoned that, if the formaldehyde concentration was close to equilibrium, improvements to Mdh activity would result in only small changes in the overall flux, whereas if the reaction was kinetically limited, these changes would manifest as large changes in methanol assimilation flux. This reasoning was confirmed by dynamic simulations (Fig. 6b). An initial attempt at varying Mdh activity by changing the level of IPTG induction was unsuccessful. Poor Mdh solubility led changes in [IPTG] to cause additional changes in growth and substrate consumption rate, which confounds the assessment of methanol flux and Ru5P availability (Supplementary Fig. 8).

Instead, we reasoned that we could use deuterated methanol ($CD_3OD$) as a chemical modulator of Mdh activity, taking advantage of the kinetic isotope effect (KIE) associated with $CD_3OD$ oxidation to specifically lower-Mdh activity in vivo without genetic intervention and the accompanying off-target effects. If cells treated with $^{13}CD_3OD$ metabolized methanol more slowly than the same cells treated with $^{13}CH_3OH$, as assessed by isotopic analysis of F6P, this would indicate the pathway was

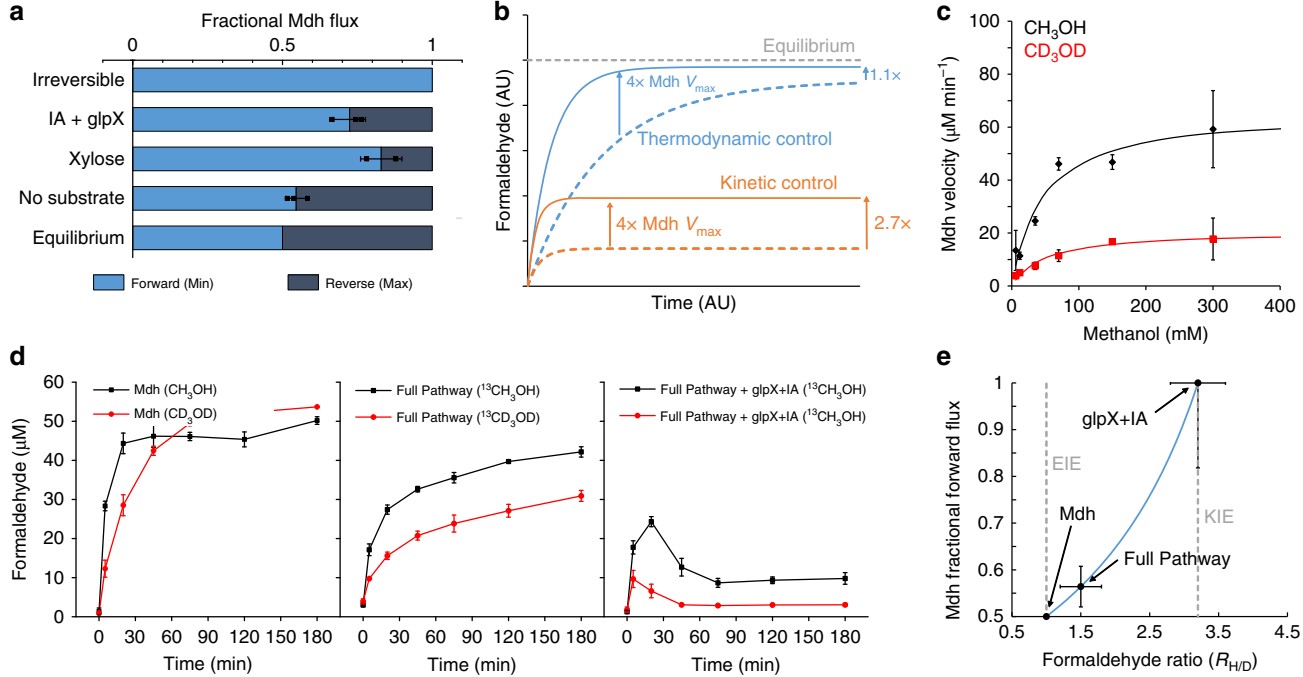

**Fig. 6** Thermodynamic and kinetic isotope effect analysis reveal Mdh kinetics limit methanol assimilation flux. **a** Upper bounds on in vivo reversibility of Mdh derived from steady-state measurements of formaldehyde in cells with Mdh or the full pathway (Mdh, Hps, and Phi), and NAD/NADH ratios. Light blue bars represent minimum fraction of Mdh flux in forward direction, and dark blue bars represent maximum fractional reverse flux. Hypothetical "Equilibrium" and "Fully Irreversible" scenarios are plotted for comparison to experimental results under various conditions. **b** Dynamic simulations of formaldehyde concentration for a hypothetical assimilation pathway where formaldehyde concentration is close to equilibrium (blue) and far from equilibrium (orange), showing the effect of increasing the $V_{max}$ of Mdh by a factor of 4 in each case. **c** Michaelis plot of Mdh activity with $CH_3OH$ (black) and $CD_3OD$ (red), highlighting the kinetic isotope effect (KIE) associated with deuterated methanol. **d** Formaldehyde concentrations over time after treatment with 250 mM $^{13}CH_3OH$ (black) or 250 mM $^{13}CD_3OD$ (red) of E. coli MG1655(DE3) $\Delta frmA$ starved cells containing various plasmids: Left, Mdh-only; Middle, full pathway (Mdh, Hps, and Phi); Right, full pathway + glpX with iodoacetate (IA) treatment. **e** The fraction of total Mdh flux in the forward direction is plotted as a function of the ratio of steady-state formaldehyde concentrations with protonated or deuterated methanol (blue line, Eq. (2)), with experimentally measured values indicated (orange circles). The minimum and maximum formaldehyde ratios are defined by the equilibrium isotope effect (EIE) and kinetic isotope effect (KIE), respectively. Error bars represent s.d. of $n = 3$ biological replicates (three individual colonies), except for **c** in which case they represent s.d. of $n = 2$ technical replicates

kinetically limited by Mdh. Further, the extent to which the measured steady-state $[CD_2O]$ was lower than the $[CH_2O]$ for the same cells could be quantitatively related to the reversibility of the Mdh-catalyzed reaction (Eq. (24)).

To implement this approach, we first verified the existence of a KIE for Mdh. Assays with purified Mdh at physiological conditions showed a roughly threefold decrease in activity between $CH_3OH$ and $CD_3OD$ across a wide range of substrate concentrations, and a $k_H/k_D$ of $3.2 \pm 1.3$ (Fig. 6c). Importantly, this KIE is significantly higher than the typical equilibrium isotope effect (EIE) for NAD-dependent primary alcohol oxidations (1.07)[27], and the secondary KIE expected for the Hps-catalyzed aldol reaction (1.04)[28], such that the impact of $CD_3OD$ treatment on methanol flux is predominantly realized through the effect on Mdh kinetics.

Repeating the in vivo formaldehyde assay with cells expressing only Mdh showed a $2.3 \pm 0.3$-fold decrease in initial formaldehyde production rate with $CD_3OD$ compared to $CH_3OH$ (Fig. 6d), verifying the in vivo utility of $CD_3OD$ as a chemical probe of Mdh kinetics. Notably, the steady-state concentration of $[CH_2O]$ and $[CD_2O]$ under these conditions were almost identical, confirming the maximum $[CH_2O]$ in vivo is constrained by equilibrium.

We next repeated the assays with cells expressing the full pathway under three conditions: (a) no supplementation, (b) glpX overexpression with IA treatment, and (c) xylose supplementation. With no supplementation, the $[CH_2O]:[CD_2O]$ ratio was

$1.5 \pm 0.3$ (Fig. 6d), suggesting a limited impact of Mdh activity on pathway flux and highlighting the thermodynamic limitation, with an estimated 56% of the Mdh flux in the forward direction (Fig. 6e). By contrast, in the glpX-IA treatment, the $[CH_2O]:[CD_2O]$ was $3.2 \pm 0.4$ (Fig. 6d), almost identical to the $k_H/k_D$, indicating that the Mdh flux was almost entirely in the forward direction (Fig. 6e).

These results are consistent with the thermodynamic calculations in Fig. 6b, and are qualitatively consistent with the isotopic analysis of F6P (Supplementary Fig. 9). In unsupplemented cells, the unlabeled (M + 0) fraction increased by a factor of 1.14 with $^{13}CD_3OD$ compared to $^{13}CH_3OH$. In the glpX-IA treatment and xylose treatment, this factor rose to 1.33 and 1.39, respectively (Fig. 6e). In the xylose case, where formaldehyde concentrations were too low to accurately determine the $[CH_2O]:[CD_2O]$ ratio, the similarity in the fractional M + 0 F6P increase to the glpX-IA case suggests methanol flux is also kinetically limited under these conditions (Fig. 6e). Taken together, these data demonstrate that, under conditions of sufficient [Ru5P], methanol assimilation flux is controlled by the activity of the Mdh enzyme.

## Discussion

In this work, we showed that in order for methanol oxidation to be forward driven, the steady-state formaldehyde concentration must remain exceptionally low (<50 μM). In carbon-starved cells,

depletion of PPP intermediates results in insufficient [Ru5P] to drive the assimilation of formaldehyde, regardless of the high activity of Hps. Under these conditions, thermodynamic calculations and experiments with deuterated methanol confirmed that Mdh flux is thermodynamically limited by proximity to equilibrium. Inhibiting glycolysis by treatment with IA, coupled with overexpression of *glpX*, activated the SBPase RuMP pathway and led to a fourfold increase in [Ru5P] and a corresponding threefold decrease in [$CH_2O$] and twofold improvement in the incorporation of $^{13}CH_3OH$. This resulted in forward-driven, kinetically limited Mdh flux, as demonstrated by the steady-state ratio of [$CH_2O$]:[$CD_2O$] in cells treated with $CH_3OH$ or $CD_3OD$.

Beyond establishing Mdh as forward-driven, the KIE experiment revealed that, under conditions of sufficient Ru5P regeneration, Mdh activity limits methanol flux, even though the evolved *C. necator* Mdh variant used in this study is the most active variant known[17,29]. This suggests that future efforts to improve methanol assimilation should focus on improving Mdh kinetics. This could be achieved through expression optimization, directed evolution, or through bioprospecting for more active candidates. Critically, our analysis reveals that kinetically improved Mdh variants will only enable improved methanol oxidation rates in cells where formaldehyde assimilation is sufficiently fast. This has important practical implications for the development of Mdh-directed evolution approaches.

Interestingly, the co-metabolism of xylose reduced initial Mdh velocity in vivo, and it was further shown that typical cellular levels of NADH inhibit Mdh by ~50%. This potentially explains why the strategy of methanol–glucose co-metabolism in *Corynebacterium glutamicum*[16], and methanol–yeast extract co-metabolism in *E. coli*[13], resulted in low rates of methanol assimilation. Genetic strategies for reducing [NADH] should therefore be explored to further enhance Mdh kinetics[30], as well developing Mdh variants that are less sensitive to NADH. Additionally, the strategy employed for Ru5P regeneration must maintain a low [NADH] in order to drive methanol oxidation.

In a recent paper, Price et al. achieved impressive rates of methanol consumption by co-localizing Mdh3 from *Bacillus methanolicus* with the Hps from *Mycobacterium gastrii*, arguing that the co-localization strategy was required to overcome the reversibility of Mdh[18]. The results presented here challenge this interpretation, as they demonstrate that [Ru5P] is the critical factor in ensuring irreversible Mdh flux. We suggest that the success of the co-localization strategy may instead be due to the very high formaldehyde $K_m$ of the *M. gastrii* Hps (2.96 mM[31], versus 147 μM for the *B. methanolicus* enzyme[32]), which requires a high [$CH_2O$] for significant activity. Increased local concentration is a hallmark of enzyme co-localization[33].

The strategy developed in this work to activate the SBPase RuMP pathway is clearly not suitable to establish a methylotrophic *E. coli* strain, as it relies on a toxic inhibitor, and expression of *glpX* leads to a futile cycle between F6P and FBP that wastes cellular energy. However, the success of this approach suggests GAPDH as a target for further metabolic engineering to improve methanol assimilation, and identifies the SBPase variant of the RuMP pathway as the preferred cycle for Ru5P regeneration. The requirement for reduced relative flux into lower glycolysis is in agreement with theoretical predictions about the stability of autocatalytic cycles such as the RuMP pathway, wherein at low concentrations of the intermediate metabolite (here Ru5P), the activity of the branching enzyme (GAPDH) must be lower than that of the autocatalytic enzyme (Hps)[34]. It is also consistent with a recent study where *E. coli* was evolved to produce sugar from $CO_2$, and where the only conserved mutation was in the enzyme that serves as the main branchpoint out of the autocatalytic Calvin cycle[35]. That both IA and *glpX* were required

for Ru5P regeneration suggests two possible mechanisms behind the improvement: IA treatment dramatically increased [G3P], and *glpX* increased [S7P]. Together, this increases the driving force for and/or the rate of the transketolase reaction that replenishes R5P and Xu5P ($\Delta G^{0'} = +3.8$ kJ mol$^{-1}$), which are converted to Ru5P. In the former case, there is a tradeoff between additional ATP cost of the SBPase pathway, and the additional driving force for Ru5P regeneration provided by its hydrolysis. In the latter case, similar improvements may be seen by using a transketolase variant with lower $K_m$ values. To avoid the F6P-FBP futile cycle, an SBPase with lower affinity for FBP could be employed.

KIEs have been widely used in the interrogation of enzyme mechanism[36], but their use in probing metabolic pathways remains limited[37,38]. To our knowledge, KIEs have not previously been used to explore limitations in an engineered pathway. Here we established a quantitative framework that leveraged KIEs to discriminate between kinetic and thermodynamic control of the Mdh reaction. The specificity of this tool was especially useful in this context, where typical genetic methods for manipulating Mdh activity caused secondary effects on growth rate that confounded the measurement and interpretation of methanol assimilation flux. This technique will be useful in future work on methanol metabolism in *E. coli*, and could in principle be extended to address limitations in other engineered pathways containing enzymes with sufficient KIEs, especially when genetic intervention proves impractical.

More generally, the chemical biology techniques used in this work (KIEs and chemical inhibition) represent useful but underutilized tools in metabolic engineering[39,40]. GAPDH is an essential enzyme in *E. coli*. Knocking it out requires deletion of both isoforms and supplementation with multiple substrates[41], and downregulating its activity requires an elaborate genetic strategy[42]. The use of a chemical inhibitor bypasses these requirements, speeding up the process of identifying rate-limiting steps, which then informs the design of genetically engineered strains to achieve the same phenotype. This strategy is complementary to the emerging strategy of designing and testing novel metabolic pathways in vitro before encoding them genetically[43–45]. Given the success of these strategies, we therefore believe that a continued close association between the fields of chemical biology and metabolic engineering will enhance our ability to rationally engineer strains in support of the goals of sustainable chemical production[46,47].

## Methods

**Reagents**. Unless specified, all chemical reagents were purchased in the highest grade available from Sigma. DIFCO M9 salts and LB, BACTO Agar, and casamino acids were purchased from BD. Methanol-free formaldehyde (16% v/v) was purchased as 1 mL ampules from ThermoFisher (Catalog #28906), and dilutions for cellular assays were prepared fresh each day. Trace Elements (MD-TMS) and Vitamin Solution (MD-VS) were purchased from ATCC. Isotopically labeled substrates were purchased in the highest grade available from Cambridge Isotope Laboratories.

**Cell culture**. LB was used in cloning and for growing cells for protein purification. For all other studies, cells were grown in M9 + medium (DIFCO M9 amended with 1% MD-TMS and MD-VS) as follows: three colonies were inoculated from fresh transformation plates into M9 + medium supplemented with the desired carbon source, 0.1% casamino acids, and the relevant antibiotics (kanamycin, 50 μg mL$^{-1}$; chloramphenicol, 25 μg mL$^{-1}$) and grown overnight. Chloramphenicol stocks were prepared in DMSO to avoid addition of Mdh substrates ethanol or isopropanol to the media. In the morning, fresh M9 + medium containing xylose or glycerol and antibiotics was inoculated at 1–2% (v/v) from the overnight cultures, grown at 37 °C, and then treated as described below for the relevant assays.

**Cloning**. All strains and plasmids generated in this study are listed in Supplementary Table 3 Heterologous genes were codon-optimized for *E. coli* and

synthesized as gBlock fragments by IDT (Supplementary Note 2). All plasmids were assembled via Gibson assembly using primers shown in Supplementary Table 4.

*E. coli* DNA fragments were amplified using Q5 Polymerase (NEB) and 0.5 µL culture as a template in a 30 µL reaction. Gene deletions (*frmA*, *talB*, and *zwf*) were generated using the CRISPR-Cas9 method described by Jiang et al.[48], with the modification of curing the pCas9 plasmid at 42 °C. The pTarget vectors were constructed in two steps: First, Gibson assembly was used to insert ~500 bp homology regions upstream and downstream of the target gene into pTargetF digested with XhoI. The homology regions were designed to make the exact deletion described in the KEIO collection[49]. Next, the N20 region for the specific gene (Supplementary Table 5) was incorporated by amplifying the vector containing the homology region with a forward primer containing this sequence (xxx-sg-F) and a common reverse primer (pTarget-sg-R), and assembling the linear fragment after DpnI digest and purification.

In generating a fully transaldolase-deficient strain, the *talA* gene was not removed completely, to minimize the risk of polar effects on expression of the primary transketolase (*tktB*) from the same operon. Instead, a mutation was made to the codon encoding the active site lysine[50] (K131) using primers listed in Supplementary Table 4. The point mutation was confirmed by sequencing of the PCR-amplified chromosomal fragment after curing of all cloning plasmids, and transaldolase activity was assessed in crude lysates as described below.

**In vivo formaldehyde timecourse assays.** In vivo formaldehyde experiments were performed essentially as described by Müller et al.[12]. Briefly, cells were grown in M9 + medium with xylose (6 g L$^{-1}$) until early exponential phase (OD$_{600}$ ~0.5), and induced with 0.1 mM IPTG. After 2 h, cells were collected by centrifugation at room temperature, washed once with M9 + medium with no carbon source, and resuspended again in carbon-free M9 + medium to an OD$_{600}$ of ~1. Various substrates were then added as described in the text, and cells were incubated for further 15 min at 37 °C before the addition of 250 mM labeled ($^{13}$CH$_3$OH, CD$_3$OD, or $^{13}$CD$_3$OD) or unlabeled methanol. Cells were then sampled periodically for internal metabolites, and supernatants were analyzed for formaldehyde, as described below.

**Formaldehyde analysis.** Formaldehyde concentration in supernatants was assayed by the Nash reaction[51] modified for 96-well plate format. Aliquot of 200 µL cells were spun down for 1 min at 16,000 × g, before transferring 125 µL supernatant to a well. The plate was kept on ice until all samples had been collected. Aliquot of 125 µL Nash reagent (5 M NH$_4$OAc, 50 mM acetylacetone) was added to each well, the plate was incubated at 37 °C for 1 h, and absorbance was read on a SpectraMax M2e spectrophotometer (Molecular Devices) at 412 nm. A standard curve was prepared fresh daily in the range from 100 to 0 µM.

**Intracellular metabolite extraction and quantification.** Intracellular metabolites were extracted and isotope labeling patterns analyzed as described in ref. [52]. Briefly, 1–2 mL of liquid culture was filtered through a 0.45 µm nylon filter, washed with 10 mL room temperature water, and then the filter was transferred into a 50 mL falcon tube containing 5 mL of extraction solution (40:40:20 acetonitrile:methanol:water) at −20 °C. After 30 min, the filter was removed, the samples were centrifuged, and the supernatants dried overnight under air. The next morning, dried metabolites were resuspended in 150 µL water, centrifuged at 16,000 × g for 20 min, and injected into the LC-MS. Labeling distributions were corrected for natural abundance using the software IsoCor[53]. For quantification of internal metabolites, the extraction solution was spiked with $^{13}$C internal standards that were previously generated by growing *E. coli* on U−$^{13}$C-labeled glucose (99%, Cambridge Isotope Labs), and area of unlabeled peaks was normalized to the area of fully labeled peak, as described in ref. [19].

**Flux balance analysis.** FBA simulations were carried out in MATLAB 2016b with the CobraToolbox (v 2.0) and the Gurobi optimization package, using the core *E. coli* model iAF1260[54], amended with the heterologous reactions catalyzed by Mdh, Hps, and Phi, and their associated metabolites. Methanol uptake was assumed to be mediated by passive diffusion. To compare flux maps across different substrates, growth was first simulated with 10 mmol gDCW$^{-1}$ h$^{-1}$ glucose, and a growth rate of 0.86 h$^{-1}$ was determined. For growth on methanol, the substrate uptake rate was varied until the growth rate was within 2% of the glucose growth rate, resulting in a rate of 42 mmol gDCW$^{-1}$ h$^{-1}$ methanol.

**Large-scale His-tag protein purification.** BL21(DE3) cells containing expression plasmids with N-terminal 6-HIS fusions of *mdh* and *glpX* were grown overnight in LB medium, then subcultured at 1% (v/v) into LB medium in baffled flasks, and grown to early exponential phase (OD ~0.5). Expression was induced with IPTG at 0.1 mM, and growth continued for 3–5 h. Cultures were centrifuged, pellets taken up in ice-cold extraction buffer (50 mM Tris-HCl (pH 7.5), 25 U mL$^{-1}$ benzonase (Sigma), 0.1 mM PMSF), and lysed by three passages through a high pressure homogenizer (EmulsiFlex-C5). The soluble fraction was separated by centrifugation at 20,000 × g for 20 min at 4 °C. HIS-tagged proteins were purified using Ni-NTA His-Bind Resin (EMD Millipore #70666) following the manufacturer's

instructions. Imidazole was removed and protein concentrated by five passages through a 3K MWCO Microsep centrifugal filter (Pall Laboratory #MCP003C41) with ice-cold Tris-HCl (20 mM, pH 7.5).

**SBPase assay.** SBPase activity was determined qualitatively by following the conversion of SBP to S7P by LC-MS/MS in the presence of purified GlpX. Since SBP is not commercially available, it was prepared in situ using purified fructose bisphosphate aldolase (FbaA), DHAP, and E4P. The assay mixture contained Tris-HCl (50 mM, pH 7.5), 1 mM each substrate, and was initiated by addition of enzyme. Products were analyzed after 1 h at 37 °C by LC-MS/MS according to the method above after 100-fold dilution in H$_2$O.

**Methanol dehydrogenase assay.** Methanol dehydrogenase activity was measured by following the methanol-dependent reduction of NAD$^+$ at 340 nm in 200 µL final volume using a clear, flat-bottom 96-well plate and SpectraMax model M2e plate reader with SoftMax Pro 6.5 software. For analysis of crude lysates, the assay consisted of 100 mM glycine-KOH (pH 9.5), 5 mM MgSO$_4$, and 1 mM NAD, and was initiated by the addition of methanol. For analysis of purified enzyme to establish KIE under physiologically relevant conditions[55], the assays consisted of 25 mM K-phosphate (pH 7.5), 10 mM Na-glutamate, 5 mM MgSO$_4$, 1 µM CaCl$_2$, 150 mM KCl, 2.6 mM NAD$^+$, and variable [MeOH] (6, 12, 35, 70, 150, and 300 mM) and were initiated by addition of enzyme (6.3 µM, final concentration). Product formation at 37 °C was quantified in comparison to a NADH standard curve. Enzyme concentration was determined by UV absorbance at 280 nm in 6 M guanidine-HCl, 25 mM Na-phosphate (pH = 6.5); $\varepsilon = 22{,}055$ M$^{-1}$ cm$^{-1}$.

**Transaldolase assay.** Transaldolase activity was determined qualitatively by following the conversion of F6P and E4P to S7P and G3P by LC-MS/MS in the presence of crude lysates of various strains. EDTA (10 mM) was added 10 min prior to starting the assay to inhibit transketolase activity. The assay mixture contained Tris-HCl (50 mM, pH 7.5), 1 mM each substrate, and was initiated by addition of enzyme. Products were analyzed after 1 h at 37 °C by LC-MS/MS according to the method above after 100-fold dilution in H$_2$O.

**Methanol dehydrogenase expression analysis.** For small-scale analysis of Mdh expression in crude lysates, 1 mL of cell culture (OD$_{600}$ ~ 0.5–1) was spun down (5 min, 16,000 × g), and the supernatant removed by aspiration. Pellets were frozen, and then lysed with 100 µL BPER-Complete at room temperature for 15 min. The soluble and insoluble fraction were separated by centrifugation at 20,000 × g for 10 min at 4 °C.

To estimate the amount of Mdh in the soluble and insoluble fraction, first the amount of Mdh in each fraction was calculated by image analysis after SDS-PAGE. Aliquot of 10 µL of the soluble fraction was mixed with Laemmli buffer (BioRad). The insoluble fraction was resuspended by vigorous pipetting in 100 µL BPER, and 10 µL mixed with Laemmli buffer. Samples were boiled for 5 min in a thermocycler, before being run on a Mini-PROTEAN TGX gel (12%, 15-well comb, BioRad) at 150 V for ~1 h. Bands were visualized with InstantBlue gel stain (Expedeon) and size was compared to BioRad Kaleidoscope pre-stained standards. Bands were identified and quantified using the software AlphaImager HP. To quantify the total amount of protein in the soluble and insoluble fraction, first the insoluble fraction was re-solubilized using 8 M guanidinium hydrochloride (Gu-HCl), and then diluted threefold in water to reduce the Gu-HCl concentration. Protein concentration was then determined by BCA Assay (Pierce). The amount of soluble Mdh was then calculated as:

$$\%MDH_s = \frac{[MDH]_s}{[MDH]_s + [MDH]_i}$$

$$= \frac{[Protein]_s(\%Protein_{s,MDH})}{[Protein]_s\left(\%Protein_{s,MDH}\right) + [Protein]_i(\%Protein_{i,MDH})} \quad (1)$$

where the subscripts s and i represent soluble and insoluble, respectively.

**Derivation of equation for determining bounds on Mdh ΔG.** Calculating a precise value of ΔG requires an accurate value of ΔG$^0$, and for the intracellular concentrations of all the reactants and products. We can reasonably assume that formaldehyde diffuses freely across the membrane, given that its appearance in the supernatant can be measured shortly after its production begins intracellularly. However, it is unlikely that the methanol concentration in the supernatant is the same as that intracellularly—at 250 mM, methanol would be 2.5-fold more concentrated than glutamate, the metabolite with highest measured concentration in *E. coli*[19]. To avoid the uncertainty in ΔG$^0$ and methanol concentration in the calculation, we use the differences in the concentrations of the relevant measurable metabolites between cells expressing only Mdh, and cells expressing the full pathway, treated in exactly the same way, to determine ΔG as follows:

$$\Delta G_1 = \Delta G^0 + RT\ln Q_1 \text{ and } \Delta G_2 = \Delta G^0 + RT\ln Q_2 \quad (2)$$

where the subscript 1 refers to Mdh-only, and subscript 2 refers to the full pathway.

Assuming that the system is at equilibrium in the Mdh-only case, then $\Delta G_1 = 0$, $\Delta G^0 = -RT\ln Q_1$, and

$$\Delta G_2 = -RT\ln Q_1 + RT\ln Q_2 = RT\ln\left(\frac{Q_2}{Q_1}\right) \tag{3}$$

Since the methanol concentration is the same for both cases:

$$\Delta G_2 = RT\ln\left[\frac{\left(\frac{[NADH]}{[NAD]}\right)_2 [CH_2O]_2}{\left(\frac{[NADH]}{[NAD]}\right)_1 [CH_2O]_1}\right] \tag{4}$$

If the NAD/NADH ratio is the same in both cases (discussed in the text), this reduces to

$$\Delta G_2 = RT\ln\left[\frac{[CH_2O]_2}{[CH_2O]_1}\right] \tag{5}$$

In reality, the system is likely not precisely at equilibrium with MDH-only, given the reactivity of formaldehyde, such that $\Delta G_1 \leq 0$. Propagating this change through the analysis leads to

$$\Delta G_2 \leq RT\ln\left[\frac{[CH_2O]_2}{[CH_2O]_1}\right] \tag{6}$$

Thus, by measuring the difference in concentration of formaldehyde, and the NAD/NADH ratio with and without Hps and Phi, we can place an upper bound on $\Delta G_2$. The relationship between $\Delta G$ and the ratio of forward/reverse flux ($J^+/J^-$) has been previously derived[56], and is

$$\Delta G = -RT\ln\left(\frac{J^+}{J^-}\right) \tag{7}$$

Therefore,

$$\Delta G_2 = -RT\ln\left(\frac{J^+}{J^-}\right) \leq RT\ln\left[\frac{[CH_2O]_2}{[CH_2O]_1}\right] \tag{8}$$

and

$$\left(\frac{J^+}{J^-}\right) \geq \frac{[CH_2O]_1}{[CH_2O]_2} \tag{9}$$

**Derivation of equation for KIE analysis.** A mass balance on formaldehyde ($F$) using mass action kinetics for each enzymatic step provides

$$\frac{d[F]}{dt} = k_1[NAD][CH_3OH] - k_{-1}[NADH][F] - k_2[Ru5P][F] \tag{10}$$

Where $k_1$ and $k_{-1}$ denote the forward and reverse rate constants for Mdh, and $k_2$ represents the irreversible forward rate constant of Hps. Solving for the steady-state concentration of formaldehyde:

$$[F]_{SS} = \frac{k_1[NAD][CH_3OH]}{k_{-1}[NADH] + k_2[Ru5P]} \tag{11}$$

Denoting all parameters for protonated methanol and deuterated methanol with "H" and "D" subscripts, respectively, and taking the ratio of steady-state formaldehyde concentrations:

$$R_{\frac{H}{D}} = \frac{[F]_{SS_H}}{[F]_{SS_D}} = \frac{k_{1H}[NAD]_H[CH_3OH]_H}{k_{-1H}[NADH]_H + k_{2H}[Ru5P]_H} \cdot \frac{k_{-1D}[NADH]_D + k_{2D}[Ru5P]_D}{k_{1H}[NAD]_D[CH_3OH]_D} \tag{12}$$

Assuming the concentration of methanol, NAD+ and Ru5P to be equal in both situations:

$$R_{\frac{H}{D}} = \frac{[F]_{SS_H}}{[F]_{SS_D}} = \frac{k_{1H}}{k_{-1H}[NADH]_H + k_{2H}[Ru5P]_H} \cdot \frac{k_{-1D}[NADH]_H + k_{2D}[Ru5P]_H}{k_{1D}} \tag{13}$$

Denoting the Mdh kinetic isotope effect (KIE) $k_{1H}/k_{1D}$ as $\alpha$, and assuming the secondary isotope effect associated with Hps to be negligible:

$$R_{\frac{H}{D}} = \frac{[F]_{SS_H}}{[F]_{SS_D}} = \alpha\left(\frac{k_{-1D}[NADH]_H + k_{2H}[Ru5P]_H}{k_{-1H}[NADH]_H + k_{2H}[Ru5P]_H}\right) \tag{14}$$

Denoting the ratio of Hps flux to reverse Mdh flux as

$$\beta = \frac{k_{2H}[Ru5P]_H}{k_{-1H}[NADH]_H} \tag{15}$$

and substituting yields

$$R_{\frac{H}{D}} = \frac{[F]_{SS_H}}{[F]_{SS_D}} = \alpha\left(\frac{k_{-1D}[NADH]_H + k_{-1H}[NADH]_H\beta}{k_{-1H}[NADH]_H(1+\beta)}\right) \tag{16}$$

Assuming [NADH] to be the same in both situations,

$$R_{\frac{H}{D}} = \frac{[F]_{SS_H}}{[F]_{SS_D}} = \alpha\left(\frac{k_{-1D} + k_{-1H}\beta}{k_{-1H}(1+\beta)}\right) \tag{17}$$

The equilibrium isotope effect (EIE), denoted $\gamma$ is defined as

$$\gamma = \left(\frac{k_{1H}}{k_{-1H}}\right)/\left(\frac{k_{1D}}{k_{-1D}}\right) = \alpha\left(\frac{k_{-1D}}{k_{-1H}}\right) \tag{18}$$

Solving for $k_{-1D}$ yields

$$k_{-1D} = \frac{\gamma k_{-1H}}{\alpha} \tag{19}$$

Substituting into equation the expression for $R_{H/D}$ yields

$$R_{\frac{H}{D}} = \frac{[F]_{SS_H}}{[F]_{SS_D}} = \alpha\left(\frac{\frac{\gamma k_{-1H}}{\alpha} + k_{-1H}\beta}{k_{-1H}(1+\beta)}\right) \tag{20}$$

Simplifying yields

$$R_{\frac{H}{D}} = \frac{[F]_{SS_H}}{[F]_{SS_D}} = \frac{\gamma + \alpha\beta}{1+\beta} \tag{21}$$

Rearranging to solve for $\beta$, the ratio of Hps flux to reverse Mdh flux, as a function of the measured formaldehyde ratios with protonated and deuterated methanol, provides

$$\beta = \frac{\gamma - R_{\frac{H}{D}}}{R_{\frac{H}{D}} - \alpha} \tag{22}$$

which is used to calculate the reversibility of Mdh at steady state,

$$\frac{MDH_F}{MDH_R} = 1 + \beta = \frac{\gamma - \alpha}{R_{\frac{H}{D}} - \alpha} \tag{23}$$

Solving for the Mdh forward flux as a fraction of the total yields

$$\frac{MDH_F}{MDH_F + MDH_R} = \frac{1+\beta}{2+\beta} = \frac{\gamma - \alpha}{\gamma + R_{\frac{H}{D}} - 2\alpha} \tag{24}$$

**Dynamic simulations of formaldehyde concentration.** The ODE describing formaldehyde production and consumption in the engineered pathway (Eq. (5)) was integrated by numerical integration in Microsoft Excel, using the kinetic parameters in Supplementary Table 6, and a time step of 0.002 (arbitrary units). The full data set is presented in Supplementary Data 1.

**Statistical methods.** Sample sizes were chosen based on precedent in the field, and no replicates were excluded from analysis. All error bars represent standard deviations calculated from the number of replicates used in the particular experiment, as indicated in the figures. Bounds on calculated ratios were derived from error propagation from the mean and standard deviation of measurements. Michaelis–Menten parameters and their confidence intervals were calculated using the MATLAB functions nlinfit() and nlparci(), respectively. Experiments were neither blind nor randomized.

**Code availability**. The MATLAB code used for calculating Michaelis–Menten parameters is provided as Supplementary Note 3. The Excel file used for dynamic formaldehyde simulations is provided as Supplementary Data 1.

**Data availability**. All data and materials that support the findings of this study are available from the corresponding author on reasonable request.

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

## Acknowledgements

This work was supported by the Advanced Research Projects Agency—Energy under the Award DE-AR0000433. B.M.W. was additionally supported by a National Science Foundation Graduate Research Fellowship under Grant No. 1122374. We also thank the Fund for Scientific Research-Flanders (FWO) for support in the form of a Ph.D. fellowship for B.V.H.

## Author contributions

B.M.W. wrote the manuscript. B.M.W. and G.S. conceived and designed the project. B.M.W., J.R.K., M.R., and B.V.H. designed and performed experiments, and analyzed data. All authors contributed to the manuscript, and read and approved the final version.

## Additional information

**Competing interests:** The authors declare no competing interests.

