## [Peer Review File · Nature Communications]

PEER REVIEW FILE

Reviewers' Comments:

Reviewer #1 (Remarks to the Author):

In this work, the authors present a study on various factors, which limit utilization of methanol as a carbon source in an already existing engineered *E. coli* strain for synthetic methylotrophy. The authors carry out an interesting set of experiments to analyze the source of the limited flux for methanol assimilation in *E. coli* in great detail.

The manuscript is well written, however, from my point of view too long. Previous work on synthetic methylotrophy was sufficiently addressed and the overall quality of the data is high. All interpretations of the data obtained and conclusions drawn appear to be valid.

I had the opportunity to review this manuscript last year and all minor comments have been addressed adequately. However, my main concern that the results obtain simply shed more light on the challenges associated with -and limitations of- this engineered biological system without any practical advance is still there. Most of the findings are quite straightforward and other scientists previously establishing this metabolic concept in microorganisms already had several of these bottlenecks in mind.

When taking a look at the remarks of the other reviewers, I see that I am not alone with this opinion. Stating that implementation of the strategies would go beyond the scope of this work is not enough, because that is exactly what I would expect for study to be published in this journal.

Furthermore, the authors made clear that they have a different opinion, but when went through recent literature associated with methylotrophy, I came to the conclusion that working on naturally evolved methylotrophs such as *Bacillus methanolicus* might be more promising. First studies conducted with this organism point in this direction.

Reviewer #2 was not able to comment on the revised version of this manuscript. We asked

Reviewer #1 to comment whether (s)he thinks the concerns of Reviewer #2 have been successfully addressed. Reviewer #1 put his/her comments in the "Remarks to Editor" section.

In summary, Reviewer #1 concerns that all limitations the authors "identified" and studied (limited Ru5P availability, unsuitable thermodynamic characteristics of the available MDHs) were already suggested by the scientists developing the first strains. (S)he thinks while it is a very detailed analysis of individual components of the already established pathways, it's hard to see "transferability" of the results.

(S)he thinks that this manuscript is not suitable for publication in "Nature Communications" as it is a detailed analysis of already known challenges, but does not represent a breakthrough.

(S)he also states that while (s)he worked on establishing "Synthetic Methylophony", (s)he do think that this is not the next big step towards a sustainable bio-based economy.

Reviewer #3 (Remarks to the Author):

I have carefully read the response of the authors to my comments and studied their followup experiments. I feel that my technical concerns are satisfied. The main issue is still whether the progress the study present is sufficient for publication in Nature Communication. While I'm still not completely sure, after re-reading the study and considering its lessons, I'm currently tend toward confirming that the paper is suitable for publication in Nature Communication. Hence I recommend accepting this study for publication.

Response to Reviewers

General Response

We thank each of the reviewers for their time in re-evaluating the revised manuscript. The concerns of the reviewers all have to do with the question of whether sufficient progress has been made on synthetic methylotrophy in this manuscript to justify publication in this journal.

We believe that our demonstration of establishing the SBPase variation of the RuMP pathway and its subsequent 3-fold improvement in the consumption rate of formaldehyde over the Tal variant, is a **major** contribution to this field. In that regard, this study is the first to successfully rationally engineer the RuMP pathway for improved methanol metabolism. All previous work on methylotrophic *E. coli* has assumed that the Tal variant would suffice for pentose phosphate regeneration. As such, a key lesson from this work is that future efforts to engineer synthetic methylotrophy should take advantage of the SBPase version of the pathway, and in this work we have provided a roadmap for the specific modifications needed to achieve this.

Beyond this specific accomplishment, we have performed a rigorous quantitative analysis of several other hypothesized major bottlenecks of methanol metabolism. To say that these ideas were already “known challenges” or that others “have these bottlenecks in mind” ignores the huge difference in value to the community between hypothesizing a challenge, and rigorously examining the validity of that hypothesis experimentally. The latter is much harder to do, but is infinitely more valuable in guiding the field in subsequent work to address those challenges. In this specific case, probing these hypotheses required us to develop several novel techniques (e.g. the use of the kinetic isotope effect to identify rate limiting reactions) that will be useful in pathway engineering beyond methanol metabolism.

Indeed, contrary to the assertion of the reviewers, our rigorous characterization *did* lead to new ideas for improving methanol metabolism: For example, we are unaware of any discussion elsewhere in the literature of the role of NADH as a kinetic inhibitor of Mdh, which we show very clearly in this work both with purified enzyme and in whole cells. This knowledge is highly “transferrable” in that it informs new approaches to improving the efficacy of this enzyme, for example by engineering an NADH-insensitive variant, or by modulating the *in vivo* NADH levels.

In a broader sense, the chemical biological tools we developed in this work represent a departure from the norms of the metabolic engineering field, and highlight the benefits of an interdisciplinary approach to pathway engineering that could potentially transform the way research is done in this area. For these reasons, which we have drawn more attention to in the revised manuscript (see specific comments below), we feel that this work is worthy of publication in this journal. We have also updated the abstract to highlight these specific contributions more clearly.

Reviewer #1 (Remarks to the Author):

In this work, the authors present a study on various factors, which limit utilization of methanol as a carbon source in an already existing engineered *E. coli* strain for synthetic methylotrophy. The authors carry out an interesting set of experiments to analyze the source of the limited flux for methanol assimilation in *E. coli* in great detail.

The manuscript is well written, however, from my point of view too long. Previous work on synthetic methylotrophy was sufficiently addressed and the overall quality of the data is high. All interpretations of the data obtained and conclusions drawn appear to be valid.

We have shortened the introduction, which we agree was quite long.

I had the opportunity to review this manuscript last year and all minor comments have been addressed adequately. However, my main concern that the results obtain simply shed more light on the challenges associated with -and limitations of- this engineered biological system without any practical advance is still

there. Most of the findings are quite straightforward and other scientists previously establishing this metabolic concept in microorganisms already had several of these bottlenecks in mind. When taking a look at the remarks of the other reviewers, I see that I am not alone with this opinion. Stating that implementation of the strategies would go beyond the scope of this work is not enough, because that is exactly what I would expect for study to be published in this journal.

Here we again draw attention to the difference between “having a bottleneck in mind”, and objectively establishing that it really is a limitation. This confirmation is needed to justify the considerable time and resource expenditure to overcome the limitation. This point has been added to the introduction:

“Such an analysis is a critical component of developing a rational strategy to improve methanol assimilation, because it justifies the considerable time and resource expenditure associated with developing the solution to the proposed bottleneck.”

Furthermore, the authors made clear that they have a different opinion, but when went through recent literature associated with methylotrophy, I came to the conclusion that working on naturally evolved methylotrophs such as *Bacillus methanolicus* might be more promising. First studies conducted with this organism point in this direction.

Genetic tools for *Bacillus methanolicus* are poorly developed, despite this being an active research area. For example, it is still not possible to make targeted knockouts in this organism, which is a critical limitation in metabolic engineering. This point has been added to the introduction.

Reviewer #2 was not able to comment on the revised version of this manuscript. We asked Reviewer #1 to comment whether (s)he thinks the concerns of Reviewer #2 have been successfully addressed. Reviewer #1 put his/her comments in the "Remarks to Editor" section.

In summary, Reviewer #1 concerns that all limitations the authors “identified” and studied (limited Ru5P availability, unsuitable thermodynamic characteristics of the available MDHs) were already suggested by the scientists developing the first strains. (S)he thinks while it is a very detailed analysis of individual components of the already established pathways, it's hard to see “transferability” of the results.

In this work, we clearly demonstrate several “transferable” results:

1) The SBPase pathway is more effective in Ru5P generation in *E. coli* than the Tal pathway. This will lead to metabolic engineering strategies focused on this pathway, which had not been discussed in the Synthetic Methylotrophy community. This point has been added to the discussion, “and identifies the SBPase variant of the RuMP pathway as the preferred cycle for Ru5P regeneration”.

2) Mdh is kinetically, not thermodynamically limited, and NADH is an inhibitor of the enzyme.

This will lead to metabolic engineering strategies for reducing cellular NADH, and/or finding variants of Mdh that are less sensitive to NADH. This point has been added to the discussion: “Genetic strategies for reducing [NADH] should therefore be explored to further enhance Mdh kinetics, as well developing Mdh variants that are less sensitive to NADH.”

(S)he thinks that this manuscript is not suitable for publication in “Nature Communications” as it is a detailed analysis of already known challenges, but does not represent a breakthrough.

(S)he also states that while (s)he worked on establishing “Synthetic Methylotrophy”, (s)he do think that this is not the next big step towards a sustainable bio-based economy.

We disagree. Several academic labs, as well as some companies (e.g. Industrial Microbes) are pursuing the goal of synthetic methylotrophy. As well as the contribution to the bio economy, establishing the complete transfer of a C1 assimilation pathway into a heterotroph would be a truly remarkable feat of metabolic engineering, with potential application to other C1 compounds, like CO₂.

Reviewer #3 (Remarks to the Author):

I have carefully read the response of the authors to my comments and studied their followup experiments.

I feel that my technical concerns are satisfied. The main issue is still whether the progress the study present is sufficient for publication in Nature Communication. While I'm still not completely sure, after re-reading the study and considering its lessons, I'm currently tend toward confirming that the paper is suitable for publication in Nature Communication. Hence I recommend accepting this study for publication

We thank the reviewer for his/her time reviewing the manuscript, and his/her recommendation.